# From In Vitro to In Vivo: A Rational Flowchart for the Selection and Characterization of Candidate Probiotic Strains in Intestinal Disorders

**DOI:** 10.3390/microorganisms11040906

**Published:** 2023-03-30

**Authors:** Flore Maillard, Maëva Meynier, Stanislas Mondot, Frederic Pepke, Chloé Galbert, Edgar Torres Maravilla, Camille Kropp, Harry Sokol, Frédéric Antonio Carvalho, Elsa Jacouton, Sophie Holowacz, Philippe Langella, Florian Chain, Rebeca Martín

**Affiliations:** 1INRAE, AgroParisTech, Micalis Institute, Université Paris-Saclay, 78350 Jouy-en-Josas, France; 2PiLeJe Laboratoire, 75015 Paris, France; 3M2iSH, UMR 1071 INSERM, University of Clermont Auvergne, INRAE USC 2018, 63001 Clermont-Ferrand, France; 4NeuroDol, UMR 1107 INSERM, University of Clermont Auvergne, 63001 Clermont-Ferrand, France; 5INSERM, Centre de Recherche Saint-Antoine, CRSA, AP-HP, Saint Antoine Hospital, Gastroenterology Department, Sorbonne University, 75012 Paris, France; 6Paris Centre for Microbiome Medicine FHU, 75571 Paris, France

**Keywords:** probiotics, characterization flow chart, IBS, IBD, lactic acid bacteria, bifidobacteria

## Abstract

Experimental and clinical evidence has demonstrated the potential of probiotic strains in the prevention or treatment of inflammatory bowel disease (IBD) and irritable bowel syndrome (IBS). However, there is little data on what the methodology leading to the identification of such strains should be. In this work, we propose a new flowchart to identify strains with probiotic potential for the management of IBS and IBD, which we tested on a collection of 39 lactic acid bacteria and Bifidobacteria strains. This flowchart included in vitro tests of immunomodulatory properties on intestinal and peripheral blood mononuclear cells (PBMCs), assessment of the barrier-strengthening effect by measuring transepithelial electric resistance (TEER) and quantification of short-chain fatty acids (SCFAs) and aryl hydrocarbon receptor (AhR) agonists produced by the strains. The in vitro results were then combined in a principal component analysis (PCA) to identify strains associated with an anti-inflammatory profile. To validate our flowchart, we tested the two most promising strains identified in the PCA in mouse models of post-infectious IBS or chemically induced colitis to mimic IBD. Our results show that this screening strategy allows the identification of strains with potential beneficial effects on colonic inflammation and colonic hypersensitivity.

## 1. Introduction

Inflammatory bowel disease (IBD) refers to ulcerative colitis and Crohn’s disease and affects 2 million Europeans and 1.5 million North Americans [1]. IBDs are characterized by inflammation of the wall of the digestive tract, increased intestinal permeability and immunological imbalance. Crohn’s disease is classically described as overactivation of the Th1 immune pathway [2], whereas ulcerative colitis seems to be more associated with the Th2 pathway [3]. The current therapeutic strategies for the management of IBD use anti-inflammatory drugs such as 5-aminosalicylic acid (5-ASA) and/or immunosuppressive drugs. Anti-tumor necrosis factor (anti-TNF) agents are the most used biological agents; however, some patients are primary non-responders to this class, and one-third will develop a loss of response over time [4]. Besides, immunosuppressive drugs are associated with side effects and a high risk for opportunistic infections [5].

Irritable Bowel Syndrome (IBS) is a functional gastrointestinal disorder affecting 4.1% of the world population [6]. IBS is characterized by abdominal pain and disturbances in bowel movements, with either diarrhea, constipation or both. The etiology of IBS is complex, involving, in some subtypes, chronic low-grade inflammation [7] and hyperpermeability [8]. A subgroup of patients developed IBS following a gastrointestinal infection (post-infectious IBS) [9]. Pharmacological treatment of IBS is currently limited to alleviating symptoms, using laxatives or anti-diarrheal drugs and antispasmodics to limit abdominal pain but does not address the root causes of IBS. 

In both IBD and IBS, intestinal dysbiosis (imbalance in the composition of the gut microbiota, with a loss of diversity) is frequently observed [10]. The gut microbiota and the immune system maintain a two-way communication throughout life, notably through the stimulation of pattern recognition receptors (PRRs) by microbe-associated molecular patterns (MAMPs) [11]. The gut microbiota also influences host physiology by producing metabolites. Short-chain fatty acids (SCFAs) produced by microbial fermentation of indigestible fiber have been shown to reduce permeability and increase mucus production, thereby improving gut barrier function [12]. The microbiota also contributes to barrier function via activation of the aryl hydrocarbon receptor (AhR), stimulating the production of IL-22 which, in turn, stimulates the proliferation of epithelial cells and the production of protective mucus and antimicrobial peptides [13]. Therefore, targeting microbiota dysbiosis is becoming a priority of biomedical research. Fecal microbial transplant (FMT) has emerged as a new strategy to cure IBD [14] and IBS [15]. However, standardization and industrialization of such a protocol are challenging and the long-term effects of FMT are still unknown. 

Probiotics are defined as “live microorganisms that, when administered in adequate amounts, confer a health benefit on the host” [16] and are currently an interesting approach used to modulate the gut microbiota in order to prevent or reduce intestinal inflammation, among others. Several strains have shown interesting functional properties in preclinical models of IBS and IBD. For example, *Liquoralilactobacillus salivarius* Ls33 displays anti-inflammatory properties [17], *Lactobacillus acidophilus* NCFM mediates analgesic functions [18] and *Lacticaseibacillus rhamnosus* LGG has beneficial effects on the intestinal barrier [19], but the effects are highly strain specific. However, systematic reviews of randomized controlled trials report little or no effect of probiotics in patients suffering from Crohn’s disease, with several well-known probiotic strains failing to fulfill the expected clinical outcome [20]. These results can probably be linked to the fact that the selection of probiotics relied mainly on their gut survival or adhesion capacity [17]. Indeed, the currently available in vitro tests are not appropriate for the identification of novel strains with beneficial effects. In 2006, the WHO/FAO expert working group on probiotics wrote: “There is a need for refinement of in vitro tests to predict the ability of probiotics to function in humans” [21]. Nevertheless, there is still little scientific literature on the combination of tests that can successfully identify probiotic strains.

In this work, we propose a new flowchart to identify strains with probiotic potential for the management of IBS and IBD and tested this on a collection of 39 lactic acid bacteria and Bifidobacteria. We combined different in vitro tests to assess the health-promoting properties of the strains tested, notably immunomodulation, epithelial barrier-strengthening effect, and the production of SCFAs and AhR agonists. The most promising strains were further tested in in vivo models of IBS and IBD. This screening strategy allowed the identification of two strains with potential beneficial effects on colonic inflammation and hypersensitivity.

## 2. Materials and Methods

### 2.1. Bacterial Growth Conditions

A set of 39 bacterial strains provided by PiLeJe Laboratoire and the French National Institute for Agriculture, Food and Environment (INRAE) were evaluated, as well as two well-studied probiotic strains, *L. acidophilus* NCFM and *L. salivarius* Ls33 (Table 1). The taxonomic assignation of each strain was verified based on the BLAST comparison of the sequence of the V3-V4 variable region of the 16S ribosomal RNA with the NCBI 16S ribosomal RNA sequence database. *Lactobacillus*, *Lacticaseibacillus*, *Lactiplantibacillus*, *Limosilactobacillus*, *Fructilactobacillus*, *Agrilactobacillus* and *Liquoralilactobacillus* strains were grown in aerobic conditions at 37 °C in vegetal De Man, Rogosa and Sharpe (MRS) broth (Biokar Diagnostics-Solabia Group, Allonne, France). *Bifidobacterium* strains were cultured in anaerobic conditions (GENbox anaer, Biomérieux, Marcy l’Etoile, France) at 37 °C in vegetal MRS supplemented with 0.1% (*w*/*v*) L-cysteine hydrochloride (Sigma, St-Louis, MO, USA). *Lactococcus* strains were grown in anaerobic conditions at 30 °C in M17 broth (Difco, Detroit, MI, USA), supplemented with 0.5% glucose and *Streptococcus* strains were grown in anaerobic conditions at 42 °C in M17 broth (Difco) supplemented with 1% lactose (Merck, Darmstadt, Germany). Bacterial growth rates were first studied during their entire growth cycle by measuring the optical density (OD_600nm_) of the cultures with a spectrophotometer (Ultrospec 10, Biochrom, Holliston, MA, USA). Bacterial counts (CFU/mL) were estimated by regular plating on Petri dishes with the corresponding media supplemented with agar 1.5% (Difco). 

For in vitro studies, bacteria were grown until 1 h post-entry into their stationary phase, washed twice with Dulbecco’s phosphate-buffered saline (D-PBS, Gibco, Waltham, MA, USA), centrifugated for 10 min at 4 °C at 9000× *g*, resuspended in D-PBS with 15% glycerol and stored at −80 °C until subsequent use. For the quantification of SCFAs and AhR agonists, bacteria were grown in 50 mL falcon tubes. Cultures were collected at 1 h post-entry into the stationary phase and centrifuged at 4000× *g* at 4 °C for 15 min. Supernatants were recovered, filtered (0.2 μm, VWR, Fontenay-sous-Bois, France) and stored at −20 °C until dosage. For in vivo experiments, cultures were lyophilized by the PiLeJe Laboratoire and resuspended in PBS at 1 × 10^9^ CFU/per mouse in 200 µL.

The *Citrobacter rodentium* strain (ATCC^®^ 51459TM DBS100) was grown overnight at 37 °C in aerobic conditions in Luria-Bertani broth (LB, Dutscher) without shaking. 

### 2.2. Cell Culture Conditions

The human colon adenocarcinoma HT-29 cell line was obtained from the American Type Culture Collection (ATCC). Cells were grown in Dulbecco’s modified Eagle’s medium (DMEM) supplemented with glutaMAX™ (Gibco), 10% (*v*/*v*) fetal bovine serum (FBS; Gibco) and 0.01% (*v*/*v*) penicillin/streptomycin (P/S, P4333; Sigma). The cultures were grown at 37 °C and 5% CO_2_ until 80% confluence was reached. 

The Caco-2 cell line was obtained from the ATCC^®^ and maintained in DMEM glutaMAX™ (Gibco), 20% heat-inactivated FBS, 1% non-essential amino acids (Gibco) and 0.01% (*v*/*v*) P/S. Cells were grown at 37 °C under conditions of 10% CO_2_ until 80% confluence was reached.

HepG2-Lucia™ AhR reporter cells were obtained from Invivogen (San Diego, CA, USA). Cells were maintained according to the manufacturer’s protocol.

Human peripheral blood mononuclear cells (PBMCs) isolated from the blood of healthy donors were obtained from Stemcells (Vancouver, BC, Canada) and stored in liquid nitrogen until use. 

### 2.3. In Vitro Immunomodulation Assays

For assessing immunomodulation of intestinal cells, HT-29 cells were co-incubated with bacteria at a multiplicity of infection (MOI) of 40 or media, with recombinant TNF-α (Peprotech, London, UK) at a final concentration of 5 ng/mL as previously described (for more details see Appendix A). Supernatants were recovered and stored at −80 °C until subsequent analysis. Interleukin-8 (IL-8) was later quantified in the supernatants using the Human IL-8 ELISA MAX Standard Set (BioLegend, San Diego, CA, USA) according to the manufacturer’s instructions. Each strain was tested 3 times from 3 independent cultures.

Immunomodulation of PBMCs was assessed in cells from five healthy donors (for more details, see Appendix A). Bacteria at an MOI of 10 or control media were added for 24 h. Supernatants were recovered and stored at −80 °C until subsequent analysis. IL-10 and IL-12p70 were later quantified in the supernatant by ELISA (Mabtech, Cincinnati, OH, USA) according to the manufacturer’s instructions. Each strain was tested 3 times from 3 independent cultures.

### 2.4. In Vitro Permeability Assay

For the assessment of the ability of strains to reinforce the intestinal barrier, transepithelial electrical resistance (TEER) was measured on Caco-2 cells, as previously described [22] (for more details, see Appendix A). Bacteria at a MOI of 40 or the media were co-incubated on the apical side of the cells. Three hours later, TNF-α (100 ng/mL final concentration, Peprotech) was added to the basolateral compartment. TEER was measured just before co-incubation and 24 h after. A ratio measuring the evolution of TEER over 24 h with the treatment, normalized by the evolution of TEER with PBS, was calculated as follows:$$\frac{TEER\;treatment\;t24/TEER\;treatment\;t0}{TEER\;control\;t24/TEER\;control\;t0}.$$

Each strain was tested 3 times from 3 independent cultures.

### 2.5. Metabolite Quantification in Bacterial Supernatants 

Production of AhR agonists in the supernatant was measured using HepG2-Lucia™ AhR reporter cells (InvivoGen, Toulouse, France). Cells were seeded into a 96-well plate and stimulated with supernatants at concentrations of 2, 10 and 20% for 24 h according to the manufacturer’s protocol. The positive control was 6-Formylindolo [3,2-b]carbazole (FICZ) at a final concentration of 18 mM. Luciferase activity was measured using a luminometer and Quanti-Luc reagent (InvivoGen). The results were normalized to the luciferase activity of the negative controls (bacterial growth medium) and cytotoxicity measurements (CytoTox 96 Non-radioactive Cytotoxicity Assay, Promega, Madison, WI, USA).

SCFA production in the supernatants was analyzed using Ottenstein and Barley’s [23] method, modified by Szylit [24]. Briefly, bacterial supernatants were deproteinized overnight at 4 °C by adding phosphotungstic acid (10% (*v*/*v*)); Sigma). Samples were then centrifuged for 15 min at 12,000× *g*. Concentrations of SCFAs were determined using a gas chromatograph (GC; Agilent 6890 N Network, Agilent Technologies, Inc., Santa Clara, CA, USA) equipped with a split–splitless injector (GC Agilent 7890B), a flame-ionization detector and a capillary column (15 m × 0.53 mm × 0.5 μm) packed with SP 1000 (Nukol; Supelco 25,236). The flow rate of hydrogen, the carrier gas, was 10 mL/min; the temperature of the injector, column and detector was 200 °C, 100 °C and 240 °C, respectively. 2-ethylbutyrate was used as an internal standard and a panel of SCFAs (Supelco, Bellefonte, PA, USA) at 10 mM was used as the technical controls. Each strain supernatant was assessed once; two technical replicates were performed for each sample. Data were processed using the OpenLab Chemsation software version 2,3 (Agilent). To determine the final SCFA concentrations, the supernatants were weighed before and after protein precipitation to obtain the appropriate multiplication factor (i.e., the supernatant to sample mass ratio).

### 2.6. Animal Studies

All animal studies were conducted in accredited research facilities and approved by local ethics committees in addition to the French government. The chemically induced colitis experiment was performed in the animal facility of the National Research Institute for Agriculture, Food and Environment (IERP, INRAe, Jouy-en-Josas, France) and was approved by the local committee on animal experimentation COMETHEA n°45 (APAFIS # 16744-201807061805486_v2). The study on infection-induced IBS was performed in the Animal Biosafety Level 2 (ABSL2) facility of the University of Clermont Auvergne (Clermont-Ferrand, France); it was approved by the local ethics committee (APAFIS # EU0116-3460) and followed the guidelines of the Committee for Research and Ethical Issues of the International Association for the Study of Pain [25].

### 2.7. Chemically Induced Colitis Study Design

Six-week-old, specific pathogen-free (SPF) C57BL/6JrJ male mice were obtained from Janvier (Le Genest Saint Isle, France). These were housed in cages of 4 in specific pathogen-free (SPF) conditions under a temperature-controlled (20 ± 2 °C) environment and a 12 h light/dark cycle with *ad libitum* access to food and water in the animal facilities of the French National Research Institute for Agriculture, Food and Environment (IERP, INRAE Jouy-en-Josas, France).

After a one-week acclimation, treatments were administered daily by intragastric gavage. Treatments were administered for 10 days before the intrarectal injection of dinitrobenzene sulfonic acid (DNBS) and continued until the end of the experiment, 3 days after DNBS injection (Figure 1).

Treatments constituted of lyophilized bacteria (PI41, PI10 or VEL12237) at 1 × 10^9^ CFU/mice in 200 μL PBS or PBS alone for control healthy mice and DNBS control mice, or 5-ASA at 100 mg/kg (Sigma) in 200 μL PBS. The PI41, PI10 and VEL12237 groups comprised 18 mice each; the control healthy group, control colitis group and 5-ASA group comprised 24 mice each. To induce colitis, mice were anesthetized with an intraperitoneal injection of a mix of ketamine (75 mg/kg, Imalgene, Boehringer Ingelheim Animal Health, Lyon, France) and xylazine (9 mg/kg, Rompun, KVP, Kiel, Germany) and injected intrarectally with DNBS (120 mg/kg of body weight, Sigma) dissolved in PBS–ethanol (70/30, *v*/*v*); control mice were injected with PBS–ethanol only. Mice were monitored daily for weight loss after intrarectal injection. At 3 days post-injection, mice were euthanized by cervical dislocation. Colons were recovered for scoring macroscopic inflammation using a mouse-adapted Wallace scale [26]. The Wallace score rates macroscopic colonic lesions according to criteria reflecting inflammation, such as hyperemia, thickening of the bowel, and extent of ulceration. In order to evaluate microscopic inflammation, sections of colons were fixed in formalin 10% (Sigma) for 24 h, then transferred into 70% ethanol, dehydrated, embedded in paraffin and stained with hematoxylin–eosin–saffron according to standard protocols at the Histology facility of @Bridge platform of UMR 1313 GABI (INRAE, Jouy-en-Josas, France). Images were analyzed with Pannoramic Viewer 1.15 software (3DHISTECH, Budapest, Hungary) and scored according to the Ameho criteria [27]. This grading on a scale from 0 to 6 considered the degree of inflammatory infiltrates, the presence of erosion, ulceration or necrosis, and the depth and surface extension of the lesions. The study was performed in two independent experiments.

### 2.8. Citrobacter Rodentium-Induced IBS Study Design

Five-week-old SPF C57BL6/J male mice were purchased from Janvier Labs and were housed in cages of 9 under conditions of 20 ± 2 °C, 12:12 h light-dark cycle with *ad libitum* access to food and water at the Animal Biosafety Level 2 (ABSL2) facility of the University of Clermont Auvergne (Clermont-Ferrand, France). Mice were orally infected with 1 × 10^9^ CFU of *C. rodentium* in 200 µL of PBS; control mice were inoculated with 200 µL of sterile PBS. From 16 days post-infection (DPI), which corresponds to the post-infectious phase, to the end of the protocol at 24 DPI, treatments were administered daily by intragastric gavage (Figure 2).

Treatments constituted of lyophilized bacteria (PI41, PI10 or VEL12237) at 1 × 10^9^ CFU/mice in 200 μL PBS or PBS alone for control healthy mice and control infected mice. Each group consisted of 18 mice. Colonic sensitivity was assessed using a non-invasive manometric method, as described previously [28]. Briefly, mice were anesthetized with isoflurane (3% in O_2_) and a balloon linked with a miniaturized pressure catheter was introduced and positioned at 1 cm from the anus. The balloon was connected to a barostat (Distender Series II, G&J Electronics Inc., Toronto, ON, Canada) providing phasic ascending pressures of 20, 40, 60, and 80 mmHg; each pressure was repeated twice. The miniaturized pressure catheter inside the colon detected the luminal pressure of the interior—IntraColonic Pressure Variations (ICV), which vary depending on the contraction of the abdominal wall—as an indirect measure of VisceroMotor Response (VMR).

Anxiety-like behavior was assessed using the Elevated Plus Maze (EPM) test (ViewPoint Behavior Technology, Lissieu, France) at 21 DPI. The apparatus consisted of two opposite open arms (37 × 6 × 0.6 cm) and two closed arms (37 × 6 × 15 cm), joined by a common central platform (15 × 15 cm) and subjected to an equal illumination (30 lux). The maze was elevated 50 cm above the floor. Mice were acclimated to the room for at least 45 min before the test. Mice were individually placed in the central zone and allowed to explore the maze for 5 min. The number of entries and time spent in closed and open arms (considered when the four paws were located within the arm) were recorded with a software-based video tracking system (EthoVison version XT13, Noldus Information Technology, Wageningen, The Netherlands). Anxiety was characterized by a low number of entries in open arms and little time spent in open arms.

### 2.9. Statistical Analyses

Principal component analysis for the in vitro results were performed with R software **4.2.3** (R Foundation for Statistical Computing, Vienna, Austria). Other statistical analyses were performed using GraphPad software (GraphPadSofware, La Jolla, CA, USA). For in vivo results, data were analyzed using non-parametric tests. For the DNBS model, a Mantel–Cox log rank test was used to study survival, a 2-way ANOVA and Dunnett’s post hoc test to study weight loss, and Kruskal–Wallis and Dunn’s post hoc test to study colic inflammation. For the infection-induced IBS model, a mixed linear model with repeated measures (repeated recordings, following increasing pressures) was used with Geisser Greenhouse correction; then, Dunnett’s post hoc multiple comparisons to study colorectal distension and the Kruskal–Wallis test followed by Dunn’s post hoc test was used to study fecal AhR agonists. A *p*-value < 0.05 was considered significant.

## 3. Results

### 3.1. In Vitro Immunomodulation Evaluation

We performed an in vitro screening on a collection of 39 bacterial strains (Table 1) as well as on two well-studied probiotic strains: *L. acidophilus* NCFM and *L. salivarius* Ls33. We quantified IL-8 secretion by HT-29 cells after stimulation with TNF-α in the presence of the strains. IL-8 is a pro-inflammatory cytokine. Therefore, bacteria enhancing its secretion are considered to have pro-inflammatory properties, while those inhibiting its secretion are considered to have anti-inflammatory properties. We observed strain-specific effects (Figure 3). The strains PI38, PI39, PI41 and LBH1072 possessed a clearly anti-inflammatory profile (−17%, −8%, −9%, −7% of IL-8 secretion, respectively, compared to the PBS control) which could be interesting for the search of strains with probiotic potential in intestinal inflammation.

We then determined the ability of the 41 strains to modulate IL-10 and IL-12p70 secretion by PBMCs. As shown in Figure 4, the 41 strains displayed diverse immunomodulatory profiles after coincubation with PBMCs. We thus identified strains that significantly induced IL-10 but little IL-12, such as strain PI10, which could be interesting strains for the management of intestinal pathologies.

### 3.2. Evaluation of the Transepithelial Electric Resistance (TEER)

We also studied the effect of the strains on barrier function. We measured the ability of the strains to limit barrier disruption following inflammatory stimulation. For each strain, we calculated the ratio of the evolution of TEER after 24 h of co-incubation and stimulation with TNF-α, normalized by the evolution of TEER induced by PBS only (Figure 5). For 24 of the strains tested, we observed a protective effect on the intestinal barrier, i.e., a less marked drop in TEER than with PBS alone. The strongest protective effects were obtained with strains PI30 and PI10.

### 3.3. Production of Metabolites

We finally quantified the production of metabolites of interest in the bacterial supernatants. The production of AhR agonists was measured in bacterial supernatants diluted to 2%, 10% and 20% in cell Hep-G2 Lucia luciferase culture medium. The concentrations of 10% and 20% resulted in high mortality of the reporter cell line, probably due to the presence of lactic acid in the bacterial supernatants. To compare all strains, we only analyzed the results obtained with 2% bacterial supernatant. At the 2% dilution, among the 39 strains in the collection, the culture supernatant of 19 significantly activated the AhR pathway. The strain with the highest AhR agonist effect was PI50, with a 3-fold change (Figure 6A). We also quantified six SCFAs in the supernatants: acetic, propionic, isobutyric, butyric, isovaleric and valeric acids. Only acetic acid was detected in biologically significant amounts (Figure 6B and Appendix A). We observed acetate production for 14 strains in the collection, including all *Bifidobacterium* in our collection (ranked in order of production PI29, PI53, PI50, PI7, PI52, LA306 and PI10).

All in vitro results were collated in a principal component analysis (PCA). This factorial method allowed us to move from a representation where each strain was a 14-dimensional vector (14 parameters studied in vitro) to a representation with 2 synthetic axes (Figure 7). We obtained a factorial plane with a horizontal explanatory axis, which summarizes 33% of the data, and a second vertical explanatory axis, summarizing 28% of the data. The two axes, therefore, represented 61% of the total inertia, so the overall quality of the analysis was considered reliable. The PCA associated strains PI10 and PI41 with anti-inflammatory properties; we therefore tested these two strains in in vivo models of chemically induced colitis and post-infectious IBS. We also added strain VEL12237 as a negative control strain because it was located in the center of the plane and did not appear to have interesting properties.

### 3.4. Evaluation of the Selected Strains in a Chemically Induced Colitis Model

We tested the PI10, PI41 and VEL12237 strains in a murine DNBS-induced colitis model. No significant difference in mortality was observed among groups, although the VEL1237 group showed the highest mortality rate of 31%, confirming its pertinence as a negative control (Figure 8A). The injection of DNBS also caused a gradual weight loss, reaching an average of 19% in the DNBS + PBS group at day 3 post-injection compared to the initial weight (Figure 8B). No preventive treatment had an improving effect compared to PBS (Dunn’s multiple comparisons not significant; *p* > 0.05).

We assessed inflammation at the macroscopic visual level by scoring colonic lesions using the Wallace scale. Inflammation was characterized by frequent large ulcers in all DNBS-treated groups, indicating very high intestinal inflammation. In this context of high inflammatory response, no treatment was able to significantly reduce macroscopic inflammation (Figure 9A); however, strain PI10 tended to reduce the score (DNBS + PBS vs. DNBS + PI10; Mann–Whitney test, *p* = 0.17). We then scored inflammation at the microscopic level using the Ameho score. According to microscopic analyses, the injection of DNBS significantly increased the level of microscopic inflammation, but we did not observe a protective effect of the treatments, probably due to the high level of inflammation achieved (Figure 9B).

### 3.5. Evaluation of the Selected Strains in a Post-Infectious IBS Model

We measured the effect of PI10, PI41 and VEL12237 strains on pain sensation by performing CRDs. Increasing pressure was applied in the colon of the mice and we recorded the variation in ICV; the greater the change in ICV, the greater the pain sensation. Infected control mice were the most sensitive to pain in this test (Figure 10). Strain PI41 was able to significantly reduce colonic hypersensitivity at 80 mmHg (Dunnett’s multiple comparisons, *p* = 0.03).

We also evaluated the effect of the selected strains on anxiety, which is frequently associated with IBS. *C. rodentium* infection increased anxiety, as measured by the time spent and the number of entries in open arms of the EPM, but we did not observe any improvement with the strains (Appendix A).

It was shown that the *C. rodentium* infection model leads to a drop in AhR activity in the feces [28]. Accordingly, we measured the AhR activity in feces collected on the day of the DCR measurements. *C. rodentium* infection significantly decreased AhR activation in the feces of infected control mice, which is consistent with the literature [28]. We observed that the PI41 and PI10 treatments significantly increased the amount of AhR agonists in the feces (Figure 11; Dunn’s post-test, *p* < 0.05).

## 4. Discussion

There is currently no consensual flowchart for the screening of probiotics, although several different methodologies have been proposed [29,30,31]. In this paper, we propose a new combination of in vitro and in vivo tests to identify new strains with probiotic potential in intestinal pathologies. We first evaluated the immunomodulatory properties of 39 strains at the local (intestinal) scale with HT-29 cells and peripherally with PBMCs. We then evaluated the ability of the strains to reinforce the intestinal barrier in vitro and their production of metabolites of interest. We combined the in vitro results to identify the most promising strains and tested them in preclinical models of IBS and IBD.

IL-8 is a key mediator associated with inflammation, and a large number of studies have shown increased levels of IL-8 in diseased mucosa [32]. Colonic IL-8 levels correlate with the macroscopic grade of local inflammation, especially in patients with ulcerative colitis, in whom large numbers of neutrophils are found in crypt abscesses [32]. There is also growing evidence that persistent low-grade inflammation plays an important role in the pathogenesis of some subsets of IBS [7]. Thus, many probiotic strains have been evaluated for their ability to modulate the production of IL-8 on the HT-29 line [33]. The strains with a pro-inflammatory profile were mainly *Lactococcus* (VEL12261, VEL12256, VEL12262), which correlated with the observations of Lammers et al. [34]. The most anti-inflammatory strain was the reference strain *L. acidophilus* NCFM (18% reduction of IL-8), which is in line with results obtained by Li et al. [35]. This test allowed us to identify several anti-inflammatory strains, among which was LBH1072, which has already been characterized as anti-inflammatory on HT-29 cells [36]. However, it should be noted that strains with a recognized probiotic effect can be strong inducers of IL-8 during co-incubation on HT-29 stimulated with TNF-α, as shown with *E. coli* Nissle 1917 [34]. This confirms that it is important to base the selection of strains on several tests modeling the various mechanisms of action of probiotics. We thus investigated the immunomodulatory properties at the peripheral level by co-incubating the strains with PBMCs to quantify IL-10 and IL-12. The important role of IL-10 for immune homeostasis in the gut has been demonstrated by the use of IL-10-deficient mice which develop chronic enterocolitis that can be prevented by the administration of IL-10 [32]. Several therapies based on the administration of IL-10 have been tested for treating IBD [37]. IL-10 was also found in lower amounts in the serum of IBS patients [38]. IL-12 is a cytokine that leads to a Th1 cell-mediated immune response, and several research programs have focused on the development of IL-12 antagonists; for example, briakinumab [39] and ustekinumab, which have been approved for the treatment of Crohn’s disease and ulcerative colitis [40]. Foligné et al. demonstrated that the IL-10/IL-12 ratio production by PBMCs incubated with the strains is a reliable indicator of the in vivo effects of chemically induced intestinal inflammation [41]. A high IL-10/IL-12 ratio is associated with a protective effect against intestinal inflammation [41]. In addition, the IL-10/IL-12 ratio produced by PBMCs from IBS patients was monitored as a biomarker in a clinical trial testing the efficacy of two probiotic strains against IBS [42]. This ratio was lower in patients than in healthy subjects at the start of the trial and increased over the course of the trial, in parallel with the alleviation of symptoms. However, we preferred to study the inductions of IL-10 and IL-12 separately because the ratio does not distinguish between strains whose inductions of IL-10 and IL-12 would be opposite and very strong and strains whose inductions of IL-10 and IL-12 would be opposite but at low levels. For the development of probiotics, we believe it was relevant to study not only the balance (or imbalance) between these two cytokines, but also their absolute quantity. In this experiment, we confirmed a strong strain-specific impact on in vitro cytokine responses of human PBMCs to bacterial strains, as observed previously [43]. For the *Bifidobacterium* strains of our collection (PI29, PI7, LA306, PI10, PI52, PI53), we observed high IL-10 and low IL-12 secretion levels. These results are in line with the studies of Latvala et al. and Neau et al., which showed that IL-10/IL-12 ratios were higher with *Bifidobacterium* strains than with *Lactobacillus* strains [44]. The immunomodulatory profiles of *L. acidophilus* NCFM and *L. salivarius* Ls-33 in co-incubation with PBMCs have already been tested by Foligne et al. [41] and Sokol et al. [45]. We obtained a similar profile (pro-inflammatory) for *L. acidophilus* NCFM, but in our experiments, *L. salivarius* Ls33 was less anti-inflammatory, maybe due to the use of vegetal media for bacterial growth in our study. The strain VEL12238 was also evaluated by Kechaou et al. as pro-inflammatory in an influenza model, with a low IL-10/IL-12 ratio and stimulating effect on the immune system [46]. Under our experimental conditions, this strain significantly induced IL-12 in two donors.

We then sought to compare the immunomodulatory profiles of the strains obtained using PBMCs with their immunomodulatory profile obtained previously using HT-29 intestinal cells. The strains with an anti-inflammatory profile with strong induction of IL-10 and no induction of IL-12 on PBMCs were PI7, LA306, PI52, PI50, PI38, PI41 and LBH1068. With the tests on HT-29 cells, the strains PI52, PI50, PI38, PI41 and LBH1068 had clearly shown an ability to reduce in vitro inflammation. In contrast, strain PI7 tended to increase intestinal inflammation, and strain LA306 had no activity on HT-29. The strains with a pro-inflammatory profile with strong induction of IL-12 but no IL-10 on PBMCs were PI30, PI3, PI9, PI27, PI43, VEL12262 and LBH1075. On HT-29 cells, strains PI43 and VEL12262 were indeed among the most inductive of IL-8. However, the other pro-inflammatory strains on PBMCs had shown near-zero activity on HT-29 intestinal cells. We can conclude that when there was a proven effect, the immunomodulatory profiles of the strains at the local and systemic levels tend to match. However, these two tests are complementary and not substitutable in our flowchart since some strains that stood out for their ability to reduce IL-8 on the intestinal line were weakly immunomodulatory on PBMCs and vice versa.

Barrier function alterations are observed in IBD and IBS patients, particularly in diarrheal and post-infectious IBS [47]. Studies have shown that the function, ultrastructure and composition of tight junctions are altered in these pathologies [48]. We therefore included TEER measurement in our screening and observed a beneficial effect on the barrier for 24 strains; for example, strain PI10 allowed an increase in barrier protection of 27% compared with PBS glycerol alone. The expression of intestinal barrier tight junction proteins can be disrupted by pathophysiological stimuli such as the presence of enteric pathogens or pro-inflammatory cytokines. Indeed, it has been demonstrated that pro-inflammatory cytokines induce the expression of the pro-pore pathway protein claudin2 on intestinal T84 cells, causing an increase in intestinal permeability [49]. We therefore compared the pro- and anti-inflammatory profiles of the strains previously identified by quantifying IL-8 on intestinal cells and IL-10 and IL-12 on PBMCs for their effect on the intestinal barrier. We were able to observe that there was not always a correlation between the immunomodulatory profile and the effect observed on intestinal permeability (TEER). On the other hand, the strains which had a deleterious effect on the intestinal permeability of Caco-2 (PI43, VEL12238 and PI54) had previously shown a very pro-inflammatory profile on the intestinal HT-29 cells.

Microbiota-derived SCFAs maintain intestinal homeostasis by regulating host immunity and inflammation [50], and alteration in fecal SCFAs have been observed in IBS [51] and IBD [52]. Butyrate has attracted considerable attention from researchers with butyrate-producing strains such as *Butyricicoccus pullicaecorum* or *Faecalibacterium prausnitzii* showing beneficial effects in IBD preclinical models [53]. Bacteria can participate indirectly in the production of butyrate in vivo by cross-feeding [54], which is why other SCFAs are relevant to quantify too. In our collection, the strains mainly produced acetate and propionate, which was expected because bacterial species known for their production of butyrate belong to the class of Clostridia [55]. We also tested the production of agonists of the AhR that control intestinal immune responses and barrier function [56]. Expression of the *AhR* gene is reduced in the intestine of patients with Crohn’s disease [57]; additionally, IBD patients produce fewer AhR agonists [58]. Lamas et al. demonstrated that oral administration of AhR agonist-producing lactobacilli to mice reduced the symptoms of chemically induced colitis [57]. Accordingly, we assumed that AhR agonist production was a relevant component in our screening strategy.

To better consider all the different parameters analyzed, we finally combined all the in vitro results in a PCA to obtain a probiotic profile of the strains. We confirmed that IL-10 production was highly correlated among the five donors; the same was true for IL-12. The acute angle formed by the production of IL-10 and acetate indicated a correlation between these two variables. On the other hand, the almost right angle formed by the IL-12 and IL-10 factors indicated that these two variables were independent of each other. Regarding the significance of the axes, we inferred that axis 2 was related to inflammation, with anti-inflammatory activity at the bottom (with IL-10 production factors, and slight acetate production) to pro-inflammatory at the top (production of IL-8). Axis 1 could represent the opposition between the production of IL-12 (left part) and the production of anti-inflammatory metabolites: IL-10 and acetate (right part). According to the analysis of the two axes, the lower right quarter of the plane seemed to be associated with anti-inflammatory properties. The most promising strains to develop probiotics for intestinal pathologies were located in this quarter, far from the origin and close to the synthetic axes. Strains PI10 and PI41 were positioned in this area of the plane and thus seemed to have a probiotic potential for the management of intestinal disorders.

To assess the relevance of our flowchart, we tested the two strains identified by the PCA as promising strains in preclinical models of IBD and IBS. The strains PI10 and PI41, plus VEL12237 as a negative control, were thus evaluated as oral preventive treatments in mice models of chemically induced colitis and infection-induced visceral hypersensitivity. The DNBS-induced colitis protocol is commonly used to model IBD [59]. Despite preparatory DNBS dose trials, our protocol (120 mg of DNBS/kg of body weight) caused more inflammation than desired in a probiotic test setting, which was evident in the high in-protocol mortality. We also recorded higher-than-expected weight loss in the control group. In this context of marked inflammation, it was difficult to observe a significant beneficial effect of the tested strains as even the chemical positive control (5-ASA) did not manage to reduce inflammation. However, strain PI10 showed a tendency to improve the macroscopic score of inflammation, with a reduction of 20% of the Wallace score compared with PBS. This strain also showed better results than the 5-ASA, which is a commonly used therapeutic agent for the induction and maintenance of remission in patients with ulcerative colitis [60]. The strain VEL12237 was associated with the highest mortality in this model, confirming its designated role as a negative control in vitro. As for the macroscopic and microscopic scores observed in this group, we did not observe a significant negative effect of VEL12237 compared to PI10 and PI41, maybe due to the bias created by the high mortality in this group. Overall, the weak response we obtained in the colitis model was underwhelming. However, we believe that sharing these results would also benefit the scientific community by preventing the waste of resources by unknowingly repeating failed studies. Indeed, the scientific literature is currently skewed toward publishing mostly positive and statistically significant results, even though null or negative findings are part of the scientific process [61]. Our results also suggested that a flowchart including additional in vitro tests measuring the anti-inflammatory potential would better predict the in vivo effects of the strains. For example, the ability of the strains to induce cytokine production upon co-incubation with dendritic cells seems to correlate with their protective proprieties in chemically induced colitis [62].

We also tested the same three strains in a *C. rodentium*-induced IBS model which causes colonic hypersensitivity and anxiety in mice [28]. Colonic hypersensitivity is estimated to occur in 60% of IBS patients [63], and multivariate studies have shown that 44% of IBS patients in consultation with gastroenterologists show an anxious state and 26% show signs of depression [64]. In our experiment, treatment with PI41 allowed a greater tolerance to colorectal distension, with a reduction of 40% of IPV variation at the highest distension (Dunn’s post hoc test, *p* = 0.04) but with no effect on behavior. Meynier et al. showed that the *C. rodentium*-induced IBS model is associated with a decrease in fecal AhR activity [28]. Indeed, the AhR regulates the differentiation of many immune cells but also the effector mechanisms of enteric neurons [65]. Although during our in vitro screening step, the strain PI41 did not produce AhR agonists, oral treatment with this strain led to a significant increase in AhR agonists in the feces. Hence, anti-inflammatory strains and strains favoring the AhR signaling pathway could resolve the low-grade inflammation and disruption of intestinal neural circuits.

## 5. Conclusions

Our results exemplify the difficulty of screening probiotic strains based on their in vitro results in tests using cellular models and quantification of metabolites of interest. Combining the in vitro results with a PCA allowed us to identify several strains associated with a potential anti-inflammatory profile. The two most promising strains were tested in murine models of IBD and post-infectious IBS. In the IBD model, a higher-than-expected inflammation did not allow us to observe any significant positive effect of the treatments, even from the chemical control 5-ASA. However, even in this context of strong inflammation, our flowchart allowed us to identify a strain tending to reduce macroscopic colonic inflammation. Concerning the IBS target, our flowchart allowed us to identify a strain able to reduce colonic hypersensitivity, which is one of the major readouts in IBS. Further research should be conducted to decipher the mechanism of action of these two candidate probiotic strains. This flowchart could also be used for other diseases or disorders since the alteration of the gut barrier and the immune system are common traits in several gastrointestinal conditions for which probiotics are a promising therapeutic complement. Overall, our results validate our flowchart as a good tool for screening bacterial collections, although it can be improved by adding tests more specifically related to the targeted diseases. No matter which in vitro tests are done, the combination of them in a single statistical analysis seems a promising strategy to better select probiotic candidates. A future perspective to improve the analysis could be the use of a more sophisticated analysis including hierarchical and ponderability factors.

## Figures and Tables

**Figure 1 microorganisms-11-00906-f001:**
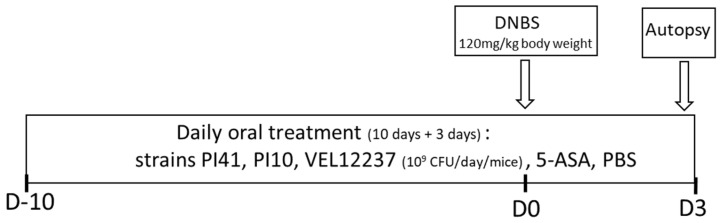
DNBS protocol. Colitis was induced by intrarectal administration of DNBS at 120 mg/kg of body weight solubilized in 30% ethanol (EtOH) in PBS. Control mice (without colitis) received only 30% EtOH in PBS.

**Figure 2 microorganisms-11-00906-f002:**
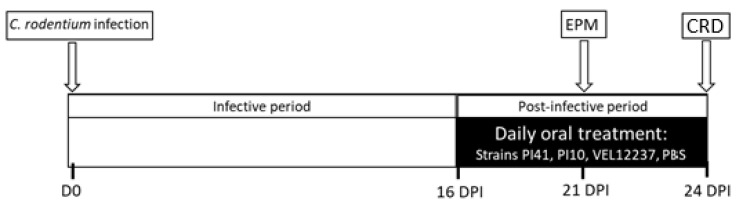
Protocol of the post infectious IBS model. DPI—days post infection; CRD—colorectal distension; EPM—Elevated Plus Maze test.

**Figure 3 microorganisms-11-00906-f003:**
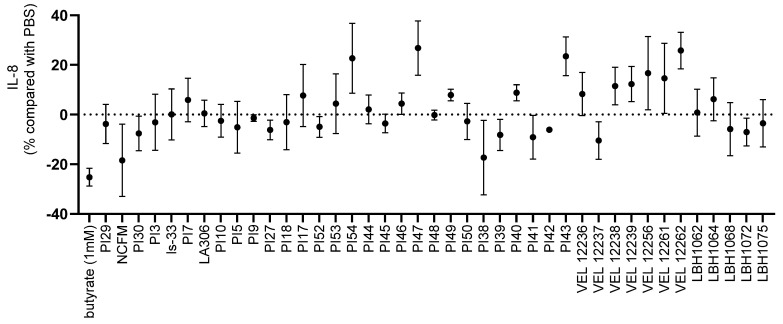
IL-8 production by TNF-α stimulated HT-29 cells in co-incubation with the tested strains compared with co-incubation with PBS glycerol only. Percentages below 0 indicate that the strain has reduced IL-8 production compared to the action of PBS glycerol and hence an anti-inflammatory potential; similarly, percentages greater than 0 indicate a pro-inflammatory potential according to this test. Each strain was tested with 3 independent cultures (3 biological replicates) in technical triplicates. Data are shown as mean ± SEM.

**Figure 4 microorganisms-11-00906-f004:**
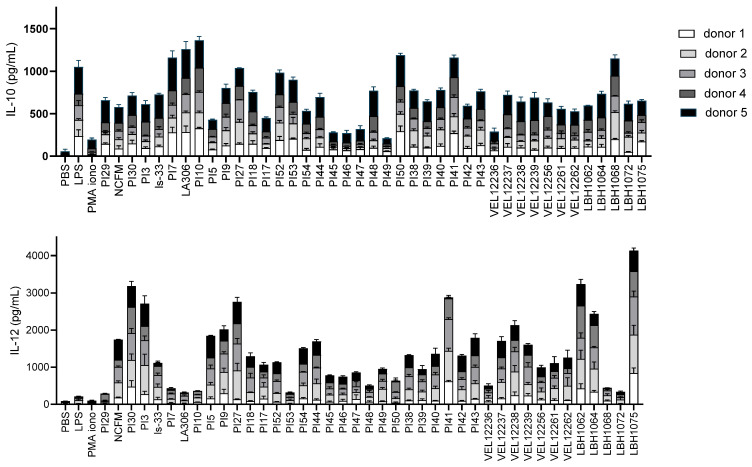
Immunomodulation of PBMCs. Strains were co-incubated with PBMCs from 5 donors. IL-10 and IL-12 were dosed in the supernatants (productions from each donor are stacked on the graph). Each strain was tested with 3 independent cultures (3 biological replicates) in technical triplicates. Data are shown as mean ± SEM.

**Figure 5 microorganisms-11-00906-f005:**
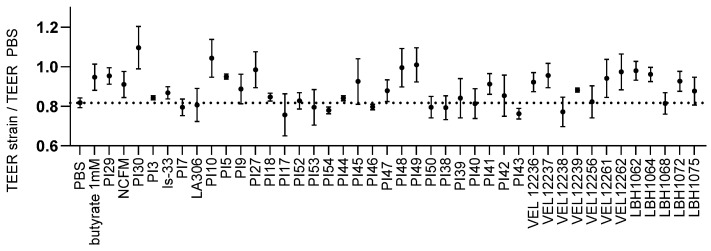
Effect on permeability in TNF-α stimulated Caco-2 cells. The evolution of TEER of the tested strain compared to the evolution of TEER with PBS only is expressed as a ratio. The control was PBS (the ratio of PBS is equal to 0.81; dotted line). A ratio above that of PBS means a protective effect on the barrier. Each strain was tested with 3 independent cultures (3 biological replicates) in technical duplicates. Data are shown as mean ± SEM.

**Figure 6 microorganisms-11-00906-f006:**
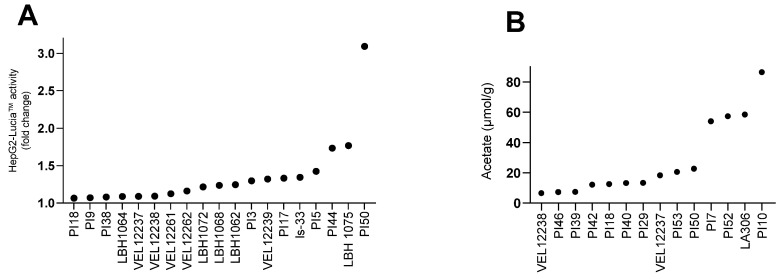
Metabolite production for bacterial strains screened in this study. (**A**) Induction of AhR activity by bacterial supernatants (only strains inducing activity >1 are represented). Supernatants were co-incubated at 2% final concentration for 24h in triplicate on HepG2-Lucia™ reporter cells. After incubation, AhR activation was assessed by determining Lucia luciferase activity in the supernatant using QUANTI-Luc™. Induction of HepG2-Lucia™ is represented as fold-change activity compared with noninduced cells (incubation with growth medium only). Results were normalized on the basis of the cytotoxicity measurement. (**B**) Acetate production in the supernatants (only strains with a net acetate production are represented). Each strain was tested once in technical duplicates. Quantification threshold for acetate: 0.1 mM.

**Figure 7 microorganisms-11-00906-f007:**
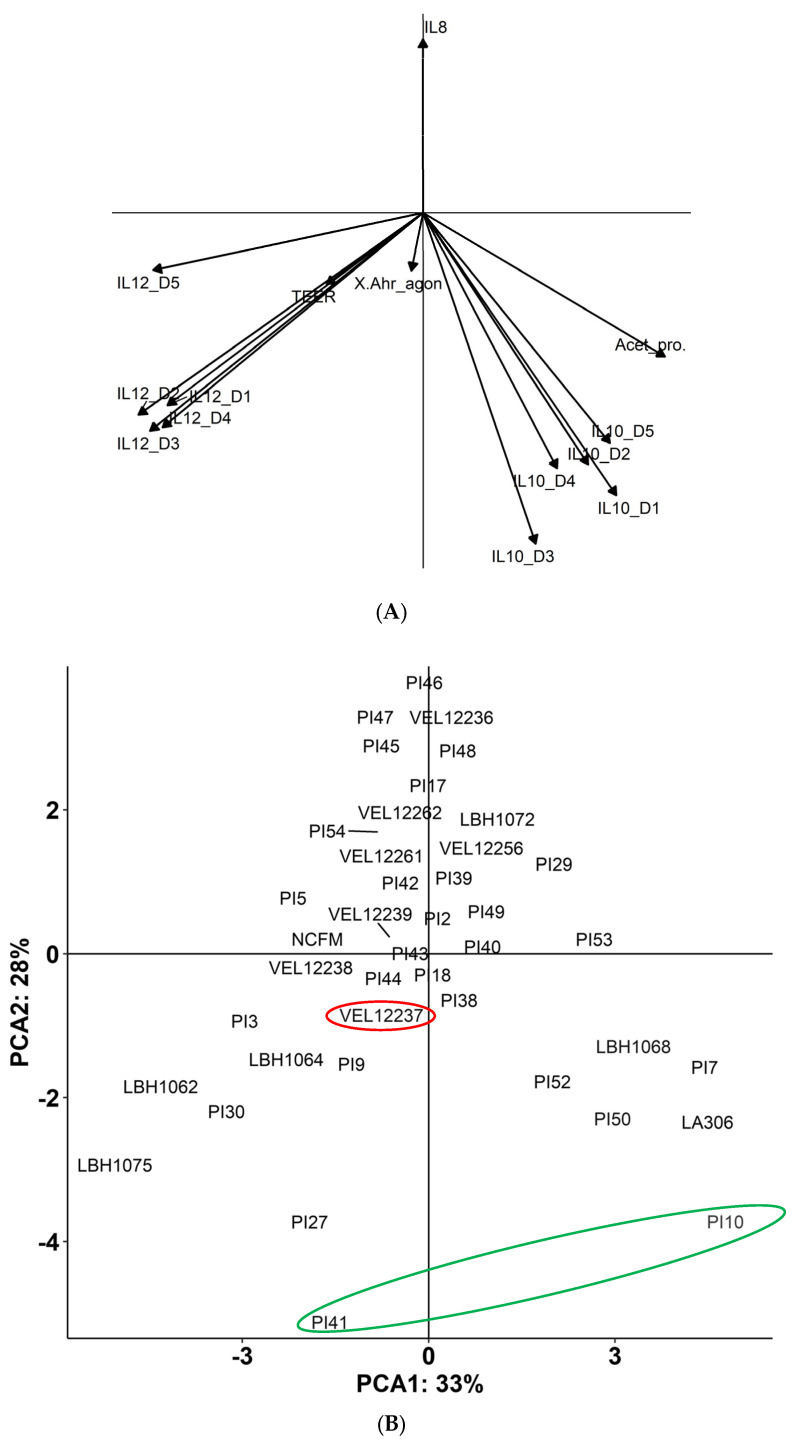
(**A**) Correlation circle and (**B**) PCA. The two axes represent 61% (28 + 33) of the total inertia. The strains circled in green (PI41 and PI10) were chosen for in vivo tests. The strain VEL12237 circled in red was selected as the negative/neutral control.

**Figure 8 microorganisms-11-00906-f008:**
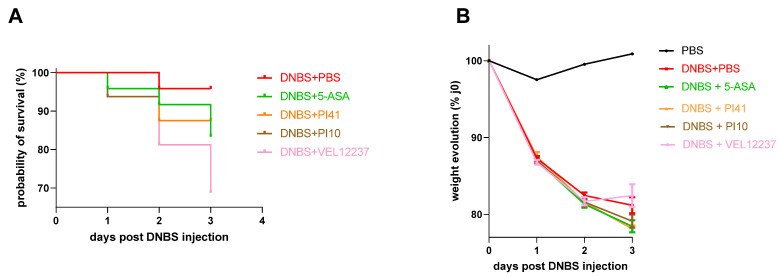
(**A**) Kaplan–Meier survival graph of the DNBS-induced colitis model. Number of dead mice post-DNBS injection: DNBS + PBS: 1/24 (4%), DNBS + 5-ASA: 4/24 (17%), DNBS + PI41: 2/16 (13%), DNBS + PI10: 2/16 (13%), DNVS + VEL12237: 5/16 (31%). No significant difference was observed in the survival curves (Mantel–Cox log-rank test *p* = 0.22). (**B**) Evolution of weight after DNBS injection (mean ± SEM). The weight loss was continuous for all the groups (for the VEL12237 group, the increase is an artifact caused by the compassionate euthanasia of the weakest mice). The number of days post-DNBS injection had more impact than the type of treatment administered (two-way analysis of variance: ‘day’ accounted for 51% of the variance, ‘treatment’ for 32% of the variance, ‘interaction between day and treatment’ for 14% of the variance). No preventive treatment had an ameliorating effect compared to PBS (Dunnett’s multiple comparisons were not significant).

**Figure 9 microorganisms-11-00906-f009:**
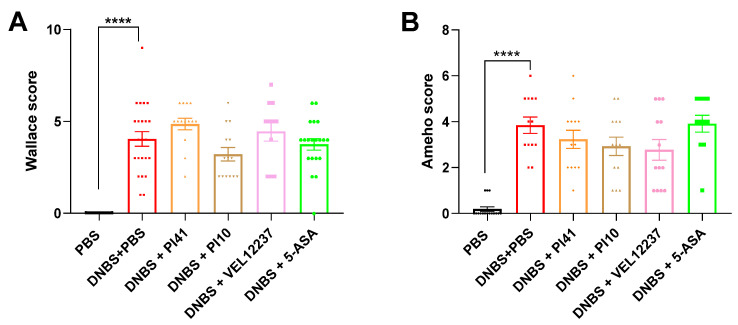
(**A**) Wallace score rating of macroscopic colonic lesions based on criteria reflecting inflammation, such as hyperemia, thickening of the bowel, and extent of ulceration; Kruskal–Wallis test, Dunn’s post hoc test. (**B**) Ameho score rating of microscopic colonic lesions based on criteria reflecting inflammatory infiltrates, presence of erosion, ulceration or necrosis, and depth and surface extension of the lesions; Kruskal–Wallis test, Dunn’s post hoc test. Data are shown as mean ± SEM. **** *p* < 0.0001.

**Figure 10 microorganisms-11-00906-f010:**
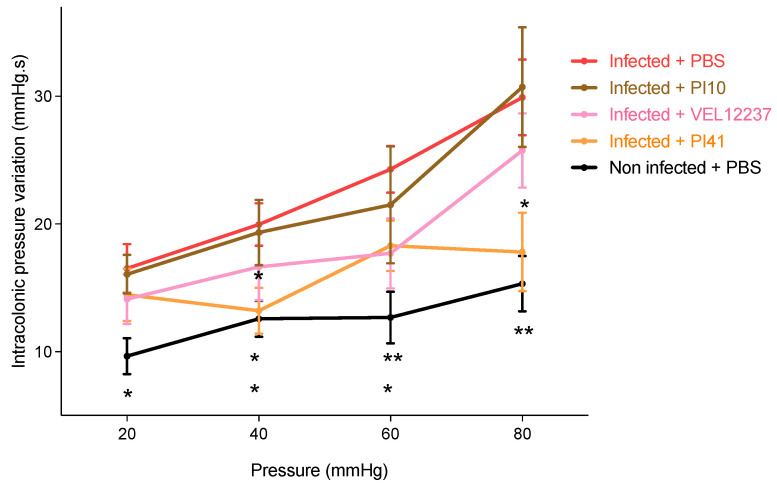
Change in intracolonic pressure following colorectal distensions. Infected + PBS, n = 17; Infected + PI10, n = 18; Infected + PI41, n = 18; Non-infected + PBS, n = 17. To analyze the data, we carried out a linear mixed model with repeated measures (repeated recordings, following increasing pressure) with Geisser Greenhouse correction; then, Dunnett’s post hoc multiple comparisons with a comparison of the means obtained for each treatment group was applied for each pressure. ‘Pressure’ factor, *p* < 0.0001; ‘treatment’ factor, *p* = 0.0002. At 40 mmHg and 80 mmHg, strains PI41 significantly limited colonic hypersensitivity compared with infected + PBS. Infected + PBS vs. Non infected + PBS; * = *p* ≤ 0.05, ** = *p* ≤ 0.01; Infected + PBS vs. Infected + PI41; *: *p* ≤ 0.05 (mean ± SEM).

**Figure 11 microorganisms-11-00906-f011:**
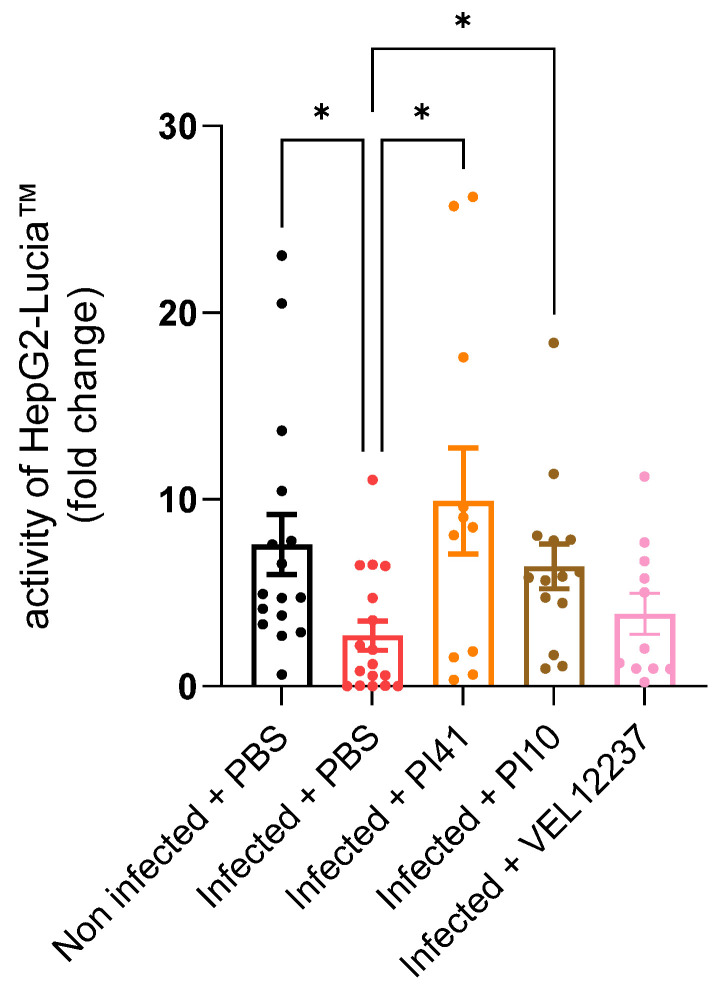
Production of AhR agonists in the feces. Lyophilized feces diluted in cell medium were co-incubated with the HepG2-Lucia™ reporter cell line. Induction of HepG2-Lucia™ is represented as fold-change compared with cell medium only. Results were normalized on the basis of luciferase activity of the negative control, cytotoxicity measurement and feces weight. Kruskal–Wallis test, *p* = 0.008, followed by Dunn’s post-test. Infected + PBS, n = 17; Infected + PI10, n = 18; Infected + PI41, n = 18; Non infected + PBS, n = 17. * = *p* ≤ 0.05. Data are shown as mean ± SEM.

**Table 1 microorganisms-11-00906-t001:** Strains included in the in vitro screening.

Bacterial Species		Strain Designation	Owner
*Bifidobacterium*	*animalis* subsp. *lactis*	PI50	PiLeJe
*Lacticaseibacillus*	*casei*	PI45	PiLeJe
*Lacticaseibacillus*	*casei*	PI46	PiLeJe
*Lacticaseibacillus*	*casei*	PI47	PiLeJe
*Lactobacillus*	*gasseri*	PI38	PiLeJe
*Lactobacillus*	*gasseri*	PI39	PiLeJe
*Lactobacillus*	*gasseri*	PI40	PiLeJe
*Lactobacillus*	*gasseri*	PI41	PiLeJe
*Lactobacillus*	*gasseri*	PI42	PiLeJe
*Lactobacillus*	*gasseri*	PI43	PiLeJe
*Lactiplantibacillus*	*plantarum*	PI44	PiLeJe
*Limosilactobacillus*	*reuteri*	PI49	PiLeJe
*Lacticaseibacillus*	*rhamnosus*	PI48	PiLeJe
*Lactobacillus*	*delbrueckii* subsp. *bulgaricus*	VEL12236	INRAE
*Lacticaseibacillus*	*paracasei* subsp. *paracasei*	VEL12237	INRAE
*Lactiplantibacillus*	*plantarum*	VEL12238	INRAE
*Lactiplantibacillus*	*plantarum*	VEL12239	INRAE
*Lactococcus*	*lactis* subsp. *cremoris*	VEL12256	INRAE
*Lactococcus*	*lactis* subsp. *cremoris*	VEL12261	INRAE
*Lactococcus*	*lactis* subsp. *lactis*	VEL12262	INRAE
*Lactiplantibacillus*	*plantarum*	LBH1062	INRAE, Escuela Nacional de Ciencias Biológicas, IPN, Departamento de Ingeniería Bioquímica, Laboratorio de Investigación en Alimentos, Distrito Federal, México
*Lactiplantibacillus*	*plantarum*	LBH1064	INRAE, Escuela Nacional de Ciencias Biológicas, IPN, Departamento de Ingeniería Bioquímica, Laboratorio de Investigación en Alimentos, Distrito Federal, México
*Fructilactobacillus*	*sanfranciscensis*	LBH1068	INRAE, Escuela Nacional de Ciencias Biológicas, IPN, Departamento de Ingeniería Bioquímica, Laboratorio de Investigación en Alimentos, Distrito Federal, México
*Agrilactobacillus*	*composti*	LBH1072	INRAE, Escuela Nacional de Ciencias Biológicas, IPN, Departamento de Ingeniería Bioquímica, Laboratorio de Investigación en Alimentos, Distrito Federal, México
*Lactiplantibacillus*	*plantarum*	LBH1075	INRAE, Escuela Nacional de Ciencias Biológicas, IPN, Departamento de Ingeniería Bioquímica, Laboratorio de Investigación en Alimentos, Distrito Federal, México
*Bifidobacterium*	*bifidum*	PI29	PiLeJe
*Lactiplantibacillus*	*plantarum*	PI30	PiLeJe
*Bifidobacterium*	*longum*	PI52	PiLeJe
*Bifidobacterium*	*bifidum*	PI53	PiLeJe
*Lactobacillus*	*gasseri*	PI54	PiLeJe
*Lactiplantibacillus*	*plantarum*	PI3	PiLeJe
*Bifidobacterium*	*animalis* subsp. *lactis*	PI7	PiLeJe
*Bifidobacterium*	*animalis* subsp. *lactis*	LA306	PiLeJe
*Bifidobacterium*	*longum*	PI10	PiLeJe
*Lactobacillus*	*helveticus*	PI5	PiLeJe
*Lactococcus*	*lactis*	PI9	PiLeJe
*Streptococcus*	*salivarius* subsp. *thermophilus*	PI27	PiLeJe
*Lacticaseibacillus*	*paracasei*	PI18	PiLeJe
*Lactobacillus*	*gasseri*	PI17	PiLeJe

## Data Availability

Raw data are available through rational demand to the authors.

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
