# Peer review of "From In Vitro to In Vivo: A Rational Flowchart for the Selection and Characterization of Candidate Probiotic Strains in Intestinal Disorders"

_microorganisms, 2023, doi:10.3390/microorganisms11040906_

Round 1

Reviewer 1 Report

THis is a scientifically sound paper well wriiten and well designed with laborious methodology.The authors aims to demonstrate the potential of probiotic strains (lactic acid bacteria and Bifidobacterium)in the prevention or treatment of inflammatory bowel disease (IBD) and irritable bowel syndrome (IBS).For this purpose, they proceeded to 'in vitro' tests for evaluating the immunomodulatory properties on intestinal and peripheral blood mononuclear cells (PBMCs), assessment of the barrier strengthening 
effect by measuring trans epithelial electric resistance (TEER) and quantification of short chain fatty acids (SCFAs) and aryl hydrocarbon receptor (AhR) agonists produced by the strains.Following that they proceed to a principal component analysis (PCA) to identify strains having
 an anti-inflammatory action.Finally, they tested most promising strains in the PCA in mouse models of post-infectious IBS or of chemically induced colitis to mimic IBD.THey develop a flowchart based on a screening strategy able to identify potential beneficial strains  in the prevention or treatment of inflammatory bowel disease (IBD) and irritable bowel syndrome (IBS).

Their experimental protocol is well designed and represented schematically for better understanding and their methodology is detailed and laborious.An extended discussion is provided based on an appropriate bibliography.

my suggestion is to ACCEPT and publish it as it should be of high interest of all scientists and industrials dealing with the potential of probiotics as medicines

Author Response

ANSWER: We thank the reviewer for the nice comments.

Reviewer 2 Report

I believe that the work has been well designed obtaining good results, even if the subject is not very new it has been reported in a sufficiently detailed and easily understandable manner for those in the field.

Author Response

(The authors gave the same response as above.)

Reviewer 3 Report

In this article, Maillard et al., showed a rational flowchart for the characterization of potential probiotic strains to improve intestinal function in gut-related disorders. It is well written, timely, and contains very interesting information for many readers, but the manuscript should go under general refinement to remove some language errors.  Moreover, to improve the manuscript authors should expand “conclusions” by briefly describing the potential advantages of using their flowchart for the selection of probiotic strains in other gut disorders. As the authors have mentioned in the conclusions section that the results of this study are not very convincing. Therefore, it could be worth mentioning the flaws of the flowchart and adding some suggestions to improve the results in the future (future directions). 

Author Response

ANSWER: We thank the reviewer for the nice comments.  IBD model results were less convincing than IBS model results not due to our selection pipeline but because of the strong inflammation achieved in our colitis model that did not allow any treatment to show positive results (even the positive control). The phrase has been modified to clarify this point. Furthermore, the conclusion section has been improved to include the reviewer’s suggestions and a general check have been performed to remove some language errors.

Reviewer 4 Report

Scientific paper is devoted to the issue that is essential for the preservation of public health, the development of an algorithm for testing probiotic strains of microorganisms to confirm their effectiveness in the treatment of intestinal disorders.

The authors have carefully reviewed the scientific data and results presented in publications in this field; the article provides a significant list of references.

In their study, the authors studied the properties and activity of 41 strains of microorganisms using validated methods in the research stage as well as mathematical processing methods.

All the data obtained are clearly presented and also compared with the results of other studies.

Overall, the article is highly relevant, scientifically sound, and advanced in this sector of microbiology.

For the successful publication of this article, I recommend making minor corrections:

Subsections 2.2 ; 2.3 ; 2.4; and 2.5 of Materials and Methods describe all the techniques used in a very detailed manner.

Such a detailed description is admirable, but very difficult to understand. It’s recommended to apply a shorter description or a schematic illustration to some subsections.

 A lot of specific acronyms are used in the text of the article - it is recommended to add a list of abbreviations at the beginning of the manuscript.

 There is a mistake in the caption for the figure 7

Figure 7 - Correlation circle and PCA. The two axes resume 61% of the total inertia. The strains circled in green were chosen for in vivo tests.

Author Response

ANSWER: We thank the reviewer for the nice comments. We have simplified the material and methods, however, as we think that it can be helpful to have a well-detailed material and methods, the complete protocols are now in supplementary data. As there is no abbreviations section in microorganisms journal, we have not included it as we are not sure if it is allowed. If the editor considers that it is necessary we will prepare it. The Figure legend 7 has been modified.